# Development of a Novel Vaginal Drug Delivery System to Control Time of Farrowing and Allow Supervision of Piglet Delivery

**DOI:** 10.3390/pharmaceutics14020340

**Published:** 2022-01-31

**Authors:** Sophia A. Ward, Roy N. Kirkwood, Kate J. Plush, Sadikalmahdi Abdella, Yunmei Song, Sanjay Garg

**Affiliations:** 1School of Animal and Veterinary Sciences, University of Adelaide, Adelaide, SA 5371, Australia; roy.kirkwood@adelaide.edu.au; 2Sunpork Group, 1/6 Eagleview Place, Eagle Farm, QLD 4009, Australia; kate.plush@sunporkfarms.com.au; 3Clinical and Health Sciences, University of South Australia, Adelaide, SA 5000, Australia; sadikalmahdi.abdella@mymail.unisa.edu.au (S.A.); May.Song@unisa.edu.au (Y.S.); Sanjay.Garg@unisa.edu.au (S.G.)

**Keywords:** sow, farrowing, cloprostenol, veterinary, vaginal deposit

## Abstract

The swine industry has evolved significantly in the recent decades, but this has come at considerable expense to piglet survival. Breeding sows for greater prolificacy has been accompanied by a greater proportion of piglets being born underweight, of lower vigor, and higher susceptibility to early mortality. Inducing sows to farrow during working hours has the potential to increase piglet survivability, but non-therapeutic injectable products are often discouraged on farms. We aimed to design and develop a novel vaginal drug delivery system (NVDDS) that could reliably trigger luteolysis and induce parturition. To achieve this, two vaginal tablets containing the luteolytic agent cloprostenol were formulated to be inserted together: one would release constituents immediately on insertion (immediate release; IR) and the other would release cloprostenol in a controlled manner (controlled release; CR). The two formulations (IR and CR) were evaluated for drug release, swelling and bio-adhesion in conditions simulating the sow vaginal environment. The IR tablet released the drug completely for 5 min whereas the CR tablet took 5 h to release 50% of the drug. Furthermore, the release kinetics were evaluated by fitting the dissolution profiles into different mathematical models. Both IR and CR tablets were best fitted by the Makoid–Banakar model which assumes release by summation of different mechanisms. The performance of the optimized formulations was studied in vivo with 161 Large White x Landrace sows of varying parity (0–5). The sows were assigned to five groups. Group 1 (SI) received a single vulval injection of cloprostenol at 0700 h (*n* = 32), group 2 (SDI) received the same dose split in two parts, at 0700h and 1300h (*n* = 33). Group 3 (IRT) animals were administered an IR tablet at 0700h (*n* = 32), while group 4 (IRCRT) received both IR and CR tablets at 0700 h (*n* = 33). Group 5 was untreated and served as a control (*n* = 32). The interval to farrowing was longer (*p* < 0.001) for controls than for treated sows, but there were no differences among cloprostenol treatments for timing of farrowing. The finding confirms the efficacy of the NVDDS for induction of farrowing in sows.

## 1. Introduction

Pre-weaning mortality continues to be an ongoing issue for the pork industry [1,2,3] with the majority of piglet loss occurring during the first 3 to 7 days of lactation [4,5,6]. In the absence of human intervention, newborn piglets are susceptible to a multitude of issues during and after farrowing including hypoxia, inadequate colostrum intake, and overlay by sows [7]. Although multiple strategies have been developed to improve early piglet survival [8,9,10,11], most are dependent on the presence of farm personnel to supervise the farrowing.

To allow for closer management of the neonatal litter, sows can be induced to farrow to provide a greater likelihood of piglet delivery during working hours. The injection of prostaglandin F2α, or an analogue (e.g., cloprostenol), can trigger regression of the corpora lutea ending progesterone secretion and stimulating piglet delivery within 22–36 h [12]. To increase the likelihood of a terminal luteolysis, treatments can be administered using the so-called split-dose protocol where the product is administered in the morning and again approximately 6 h later [13]. Using the split-dose protocol, the proportion of sows farrowing the following working day increased from 56% to 84% [13]. A graphical explanation of this process is presented in Figure 1.

Provided the treatment is given within the final two days of the herd-specific due date, induction of parturition is a safe and effective measure for improving piglet survival [7,8,15]. However, in commercial practice, a major constraint in the use of induction is the need for injectable treatments that can be acutely painful and stressful to sows [15]. For this reason, many commercial farms are against the use of injectables for non-medicinal purposes, which limits the potential for luteolytic agents in production.

The present study describes the development of novel vaginal drug delivery systems (NVDDS) containing the synthetic prostaglandin, cloprostenol. The NVDDS was designed to control the release of cloprostenol after tablet placement in the sow vagina. Super-disintegrants are often included in IR tablets to achieve faster disintegration [16]. For the controlled release (CR), the drug release rate from the tablet can be controlled over time using slow-release polymers either alone or in combination. A hypothetical result for IR and CR NVDDS is presented in Figure 2. 

To achieve the desired release profile of cloprostenol from CR, hydroxypropyl methylcellulose (HPMC) can be used for its biodegradability and non-toxic properties [17,18]. This cellulose-based polymer will swell and form a gel-like matrix to control release of an incorporated drug into the surrounding fluid. Different viscosity grades and quantities of HPMC will determine how well the polymer will relax with volume expansion, allowing cloprostenol, a highly water-soluble drug, to diffuse out of the tablet matrix [19,20]. The CR NVDDS must not only release optimally but should be mucoadhesive while the drug is releasing to ensure retention. Different concentrations and gradients of HPMC and lactose should be compared to identify an optimal formulation. 

Previous reports determined that a 50% dose [13,21] of prostaglandin or its analogue injected into the vulva was as effective as an intramuscular injection at the full label dosage. Therefore, in the present study, NVDDS was formulated to deliver relatively low cloprostenol into the vagina either immediately after insertion (IR) or to release in a controlled manner over a sustained period (CR). Our hypothesis was that the double administration of cloprostenol via IR and CR NVDDS will be just as effective at inducing sow parturition as the traditional ‘split dose’ [12,14] intravulval delivery protocol. 

## 2. Materials and Methods

### 2.1. High Performance Liquid Chromatography (HPLC)

Using the method outlined by Kalikova et al. [22], enantiomers were separated and quantified. The separation system consisted of a Lux Cellulose-1 column (Phenomenex, Pty, Ltd., Lane Cove, NSW Australia) acetonitrile-sodium dihydrogenphosphate (pH 3; 20 mM) (1:2, *v*/*v*) as the mobile phase. HPLC was performed using a Shimadzu LC system (Shimazu Corporation, Kyoto, Japan) with a column temperature set to 20 °C, a flow rate of 0.7 mL/min, and a wavelength 274 nm. Retention time of cloprostenol was 6.5 min (linearity range: 2.5–30 µg/mL; *r^2^* = 0.99) as presented in Figure 3.

### 2.2. NVDDS Preparation 

The IR tablet was formulated to release constituents immediately on contact with biological fluid and be sturdy enough to resist tablet breakage prior to use. Kiccolate ND-2HS (5.4 mg) was used as a super disintegrant, Magnesium stearate (1.8 mg) and Aerosil 200 (1.8 mg) were added for improving flowability, and Microcrystalline cellulose KG-802 (19 mg) was used for tablet compressibility. Mannitol (60 mg) and anhydrous lactose (85.5 mg) were used as fillers to obtain the desired final weight of each tablet (186 mg). These compounds were shaken thoroughly before a racemic mixture of cloprostenol (125 µg) was ground into the mixture. To ensure even drug dispersion, tablet mix was incorporated with cloprostenol in small amounts and shaken thoroughly between each addition. For the CR tablet, cloprostenol (125 µg) was combined with Magnesium stearate (1.8 mg), Aerosil 200 (1.8 mg), Microcrystalline cellulose PH-102 (19 mg) and Mannitol (60 mg). Between CR formulations, different viscosity grades of HPMC and ratios of HPMC and anhydrous lactose were tested, as presented in Table 1.

Once all the tablet ingredients were thoroughly mixed in a polythene bag, the tablets were prepared by direct compression method.

For each formulation, ten tablets were weighed, measured for thickness using a vernier caliper (Copley Scientific, Colwick, Nottingham, UK) and hardness tested using a digital force gauge (Electrolab model EH-01P; Cupertino, CA, USA) for peak breaking point (KPI).

### 2.3. Friability Test

For the selected formulations IR and CR6, tablets (*n* = 35) were randomly selected from the batch, dedusted, and weighed together. Tablets were then placed in a dual drum tablet friability tester (EF-2 Friabilator USP, Electrolab Pty, Ltd., Goregaon East, Mumbai, India) rotated 100 times and dedusted prior to reweighing. The difference in weight before and after was expressed as the percentage lost.

### 2.4. Drug Content Uniformity

For the selected formulations IR and CR6, a sample of tablets (*n* = 10) was taken from the batch. Each tablet was dissolved in a 100 mL volumetric flask of Milli-Q water and assessed for total drug content using HPLC. The acceptance value was calculated using the formula:(*M* − *X*) + *k* × *s*
where *M* = reference value, *X* = mean of individual contents, *k* = acceptability constant (2.4), and *s* = standard deviation.

### 2.5. Dissolution Study

To assess the release behaviour of the developed formulations, each tablet was placed into a vial containing 10 mL of Milli-Q water and the vials were rotated at 25 rpm at 37 ± 1 °C [23]. One millilitre of samples was withdrawn at each sample time. The medium was kept at a constant volume by refilling it with fresh water. The withdrawn samples subsequently were filtered through a 0.45 µm filter and assayed by HPLC.

Release studies were conducted on each formulation with three tablets and average values were plotted against time. 

### 2.6. Polymer Swelling for CR NVDDS 

To assess tablet bio-effectiveness in vitro, the artificial vaginal solution was formulated to simulate the pH and temperature of sow vaginal secretions in late gestation (Appendix B). Simulated porcine vaginal fluid (PSVF) was prepared using the composition reported by Owen and Katz for humans [24] and adjusted with NaOH to a pH of 7. The pH was adjusted to suit the vaginal conditions of Large White x Landrace sows in late gestation (Appendix B) as well as reports on porcine vaginal environment by Lorenzen et al. [25]. 

The rate of medium uptake for each tablet formulation was assessed using the method of Chaibva et al. [26]. Dry tablets were weighed using an electronic balance and fixed onto a pre-weighed plastic square. The tablets were covered with 10 mL of PSVF and rotated at 10 rpm at 37 ± 5 °C for ten hours. Every hour, tablets were removed from PSVF, blotted lightly with kimtech tissue paper, and weighed. Swelling was assessed using the following equation:$$\mathrm{Swelling}\;\left(\%\right)=\left\{\left(W_{t}-W_{o}\right)/W_{o}\right\}\times 100$$
where *W_o_* = dry tablet weight and *W_t_* = swollen tablet weight.

### 2.7. CR NVDDS Bio-Adhesion Test

Porcine vaginal tissue was obtained from a local slaughterhouse (Murray Bridge, SA, Australia) and transported in PSVF (pH = 7) on ice to the laboratory. The vaginal tissue was removed from surrounding tissue, rinsed three times with isotonic saline solution [27], and stored in aluminium foil at −20 °C for later analysis as described by Hiorth et al. [28].

CR tablets were fixed onto a stainless-steel probe with cyanoacrylate adhesive on a hydraulic press. Porcine mucosa was thawed in PSVF (pH 7) at 37 ± 1 °C for 60 min using a magnetic stirrer and heated disk [28] and secured into the mucoadhesion rig suspended in PSVF at 37 °C ± 1 °C. The TA.XTplus texture analyser (Arrow Scientific, Pty, Ltd, Gladesville, NSW, Australia) evaluated bio-adhesion force, with a schematic diagram presented in Figure 4.

Following the method of Hiorth et al. [28] for the evaluation of vaginal tablets, the probe with the attached tablet was moved down to the tissue at 1 mm/s until contacting the vaginal mucosa, applying a contact force of 5.0 g for 30 s. The probe was then separated at a speed of 0.1 mm/s until the tablet was detached (*Fmax*). Each batch was evaluated in triplicate.

### 2.8. In Vivo Testing of NVDDS

After assessing the activity of selected formulations, the chosen CR (CR6) and the IR tablets were tested in vivo using sows housed at the University of Adelaide Roseworthy Piggery with approval from the institutional ethics committee (AECS09:34706). Approximately seven days before farrowing, Large White x Landrace sows (*n* = 161) were moved into individual farrowing crates where they remained for the duration of the trial. Sows had free access to fresh water and were fed twice daily with a standard pelleted diet formulated to meet all nutrient requirements. Two days before their expected due date (day 113 of gestation), sows were assigned to one of five treatments: SI, SDI, IRT, IRCRT or control.
SI: Injection of 250 µg cloprostenol (Juramate^®^, Jurox Pty, Ltd., Rutherford, NSW, Australia) into the vulva at 7:00 h;SDI: Injection of 125 µg cloprostenol into the vulva at 7:00 h and again at 13:00 h;IRT: Insertion of IR tablet at 7:00 h and again at 13:00 h; for each vaginal deposition, the tablet applicator was sanitised (F10SC Veterinary Disinfectant), rinsed with water and lubricated (Obstetrical Lubricant, ZebraVet, Sherwood, QLD, Australia)IRCRT: Insertion of IR and CR (formulation CR6) tablets at 7:00 h.Control: No cloprostenol administration

Data recorded were the interval from treatment administration to the delivery of the first piglet (min), total born litter size, piglet birthweights and piglet pre-weaning mortality.

### 2.9. Statistics 

A one-way ANOVA was used to analyse in vitro data using IBM SPSS v20 statistical software package and data are presented as the mean ± standard deviation of the mean. Confidence limit was set at 95% (*p* < 0.05). In vivo sow data were assessed using a general linear mixed model with the random terms room (identical farrowing rooms 1–5) Sow ID and farrowing batch (April, May or June 2021) and fixed effects parity group (P0–P5) and treatment.

## 3. Results 

### 3.1. Tablet Physical Properties 

IR and CR tablet thickness and hardness results are presented in Table 2.

### 3.2. Dissolution Studies

#### 3.2.1. IR-NVDDS

The IR-NVDDS disintegrated within 5 min of contact with the surrounding fluid, with complete release of cloprostenol (Figure 5). Incorporating the super-disintegrant, Kiccolate ND-2HS, provided instantaneous disintegration [29] to enhance the drug dissolution rate.

#### 3.2.2. CR-NVDDS

The CR tablet formulations released cloprostenol gradually over a six or eight-hour period, with dissolution profiles presented for the formulations CR1–CR6 (Figure 6). Formulations CR1 and CR2 had the highest viscosity grade (K100) of all formulations but the lowest proportion of HPMC in each tablet. Combining lower viscosity grades (K15 and K4) with double the amount of HPMC (CR3) resulted in a slower release after the fourth hour. For CR5 and CR6, incorporating E50 with K15 and K4, respectively, produced optimal release of cloprostenol at the desired rate with ~50% release by the fifth hour. CR6 achieved the target release profile and was selected for the in vivo study. 

### 3.3. Mechanism of Drug Release 

Mathematical models are employed to better understand the mechanism of drug release from dosage forms. The mathematical model that best fits the dissolution data helps to predict the release kinetics of the drug. To understand the mechanism of cloprostenol release from the IR and CR6 tablet formulations, several mathematical models were employed to fit the experimental data to the theoretical curve (see Appendix A). The data from the dissolution studies conducted were fitted into the individual kinetic models and the goodness of fit of the experimental release with predicted release profile was evaluated using three common statistical criteria in combination; the adjusted *R^2^*, the RMSE and the AIC.

The release profile of cloprostenol from IR tablets was fitted by first-order, Hopfenberg, Peppas–Sahlin and Makoid–Banakar release kinetics, with an R^2^ value of 1.00. First-order release kinetics describe the drug release rate from the pharmaceutical dosage form as being proportional to the amount of drug remaining in its interior. The amount of drug released decreases by a unit of time. The Makoid–Banakar release model, on the other hand, assumes total drug release is the result of several mechanisms such as burst release, controlled and diffusional release among others. Hopfenberg assumes that the rate-limiting step of drug release is the erosion of the matrix itself while, in Peppas–Sahlin, the drug release is controlled by both Fickian diffusion and case II relaxations. Comparing the models using other statistical criteria, Makoid–Banakar perfectly fits the release of the drug from the IR tablets. The release of the drug from the IR tablet, in our case, could be due to a burst release. Release of the drug from the CR6 tablet was best described by a Makoid–Banakar release model with an adjusted *R^2^* value of 0.9776 (Figure 7).

### 3.4. Swelling Tests

The swelling profiles for NVDDS formulations CR1–CR6 are shown in Figure 8. Formulation CR1 exhibited non-uniformity between tablets, indicated by large deviation bars. After the eighth hour, the tablet weights decreased as the matrix eroded. Formulations CR5 and CR6 showed similar swelling profiles that gradually increased from hours one to eight and then declined in the hours following as the formulation started to erode. 

The visual differences between formulations containing a low concentration of HPMC (CR1), and the desired formulation containing 26% HPMC (CR6), are presented in Figure 9. CR6 exhibited even swelling within the triplicate across hours 1, 5 and 10 (Figure 9A). For CR1, physical differences were observed between the tablet triplicates with uneven tablet swelling and erosion (Figure 9B).

### 3.5. Bio-Adhesion Tests

Following the method outlined by Hiorth et al. [28] for vaginal drug delivery in women, we determined the maximum detachment force for testing CR formulations. The values for CR2–CR6 (Figure 10) were similar to those reported in the study for HPMC based tablets (0.14 ± 0.9 *Fmax* [N]). CR3 and CR6 had the highest mean detachment forces but were still within the values reported by Hiorth et al. [28]. Formulation CR1 had a significantly lower detachment force than other formulations (0.06 ± 0.01 *Fmax* [N]) and a lower value than what was reported by Hiorth et al. [28].

### 3.6. Selected Formulations Uniformity and Friability Tests

From the results of previous in vitro testing, NVDDS formulation CR6 was selected to test in sows along with the IR NVDDS. The formulated tablet batches selected for in vivo study were assessed for tablet uniformity with test results presented in Table 3. The content uniformity scores for selected tablets (*n* = 10) fit within the maximum allowed acceptance value (L1 ≤ 15 for solid dosage tablets). 

Friability testing was also conducted on selected formulations with no physical signs of cracking, cleaved or broken tablets after 100 rotations in the tablet friabilator. The percentage of tablet loss (*n* = 35 tablets) must fit within the maximum acceptance value (percentage lost ≤ 1%). Values are presented in Table 4. 

### 3.7. In Vivo Tests

There were no differences among treatments for sow parity, mean litter size, average piglet birthweight or preweaning survival rate, as presented in Table 5. 

All sows receiving cloprostenol, regardless of dose or route of administration, had shorter intervals to farrowing than the control sows (*p* = 0.001). The average times to farrow after treatment administration were all within the 23–32 h expected for successful farrowing induction [14]. No significant differences were observed between sows that were induced by injection or IM and CR tablets (Figure 11). Sows that received a single injection of cloprostenol had the same induction success rate as those that received a split dose.

## 4. Discussion

The dissolution test results of the IR tablet demonstrated the formulation achieved the rapid release of cloprostenol. The super-disintegrant (Kiccolate ND-2HS) resulted in the rapid and complete breakdown of the tablet when hydrated by aqueous media (PSVF). Adding microcrystalline cellulose (KG-802) into the IR formulation strengthened the tablet at lower compaction forces [30] and provided a sturdy tablet that could still break down rapidly in biological fluid. The developed formulations also resisted chipping or capping after 100 spins in the tablet friabilator; not obviously cracked, cleaved, or broken tablets were present in the tablet testing samples after tumbling. 

The drug release profiles of CR tablet formulations showed both the viscosity grade and quantity of HPMC polymer contributed to tablet release properties. CR3 had a lower HPMC viscosity grade than CR1 and CR2, but a higher concentration of HPMC, resulting in a slower release after the fourth hour. Similar results have been reported in other studies [31,32] with higher concentrations and viscosity grade slowing down the release of water-soluble actives. Increasing the concentration of HPMC allows more molecules to pack together and form a matrix with greater viscosity [32]. Reducing the HPMC viscosity grade and keeping the higher concentrations of HPMC provided an even release of cloprostenol into the dissolution fluid every hour. Similarly, Vueba et al. [33] found the combination of high and low HPMC viscosity grades increased gel heterogeneity and provided stable uptake of fluid. Finer HPMC particles allow rapid and uniform hydration of HPMC molecules and a more controlled drug release over time [34].

With the lowest concentration and highest viscosity grade of HPMC, formulation CR1 presented an uneven release of cloprostenol and an unreliable swelling profile between tablet samples. The concentration of HPMC in this formulation (10%) was lower than the ‘percolation threshold’ reported in other studies [31,35] and may have contributed to the difference in swelling percentage observed in Figure 9B. Although HPMC concentrations as low as 10% (*w*/*v*) have been reported [18,36,37,38], low levels of HPMC and high lactose content can be associated with sudden changes in matrix integrity and inconsistent results [38,39]. Substances with higher particle size require higher concentrations to percolate the tablet [39,40], so combining with a lower viscosity grade may improve tablet robustness. In Figure 8, formulations CR1 and CR2 showed signs of erosion after the eighth hour yet CR3 retained a strong gel matrix with no signs of core erosion. At the surface of the gel tablet layer, formulations containing greater concentrations of HPMC are less susceptible to erosion as there are more polymetric chains to withstand forces surrounding the gel [37]. Concentrations of HPMC below the critical polymer concentration cannot withstand the shear forces surrounding the gel for as long, and erosion occurs at a faster rate [41]. 

In addition to having an unstable release, CR1 also had the weakest detachment force. CR6 and CR3 had higher HPMC concentrations (>24%) and the higher concentrations have also been found to increase mucoadhesive strength in previous studies [37,42]. Increasing the concentration of HPMC in tablet formulations increases the likelihood of chain entanglement between the hydroxyl groups and amide groups of the mucin layer [38]. 

The optimized formulation for controlled release of cloprostenol was CR6, containing both high and low viscosity grades of HPMC (E and K) and a higher concentration of HPMC (26%, *w*/*v*). The results of in vitro testing indicate that a combination of viscosity grades and a concentration within reported percolation thresholds for HPMC are important for a reliable, even release rate of cloprostenol. 

In vivo studies with periparturient sows indicated a successful release of cloprostenol from tablets and diffusion through vaginal mucosa to initiate luteolysis. No differences were observed between single and double injection of cloprostenol, indicating the treated sow herd did not experience incomplete luteolysis. 

The insertion of the IR NVDDS was able to successfully induce farrowing 24–34 h after application. At the very least, the IR tablet was able to disintegrate and release cloprostenol, and enough of this cloprostenol permeated through the vaginal mucosa for luteolysis to be triggered. Whether the CR6 tablet helped to prevent incomplete luteolysis during this period requires further testing on larger sow herds. 

The induction of parturition had no effect on birthweight, piglets born alive or percentage of preweaning survival. The results are similar to those previously reported for the survival of the litter for sows induced two days before an intended due date [43]. Straw et al. [39] found lighter birthweights for induced sows but similar weights by day 12. Conversely, Gunvaldsen et al. [38] reported similar birthweights between induced and non-induced litters, but found average daily gain was lower for piglets born to induced sows [44]. Further investigation into the growth rate of piglets vs the number of days in gestation is recommended to understand the possible side effects of implementing an induction protocol in commercial production. 

## 5. Conclusions

Vaginal formulations offer the potential for a non-injectable route for cloprostenol to successfully induce sows to farrow during the working day. The immediate-release NVDDS formulation was able to disintegrate and release 100% of active constituents within 5 min and tablets did not show obvious signs of being cracked, cleaved or broken in friability tests. Formulating a sustained release NVDDS with 26% HPMC (*w*/*v*) and a combination of polymers K15 and E50 produced a tablet with a desirable, even release of cloprostenol, resistant to early erosion, tablet uniformity and appropriate bio-adhesion in vitro. As the sample of sows used in this study did not show signs of incomplete luteolysis, it is not evident how effective IR and CR tablets are for reliably inducing farrowing over a single IR dose. Future investigations should test the effectiveness of IR and CR NVDDS on larger sow populations, particularly where there is evidence of single-dose administrations of cloprostenol being not as reliable for inducing sows over the split-dose method.

## Figures and Tables

**Figure 1 pharmaceutics-14-00340-f001:**
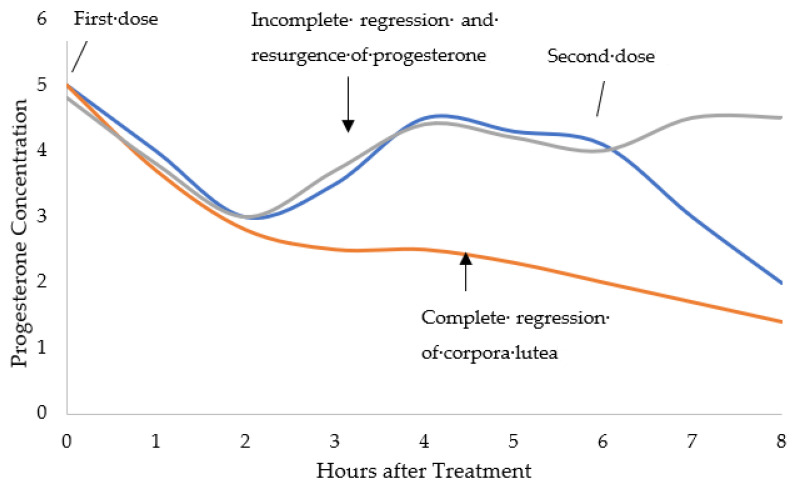
Graphical representation of how progesterone concentrations fluctuate when sows are given a luteolytic agent (i.e., cloprostenol). The first dose can trigger complete regression for successful induction (orange line) or incomplete regression causing an unsuccessful induction (grey line). Administering a second luteolytic dose 6 h after the first should trigger complete regression and result in a successful induction (blue line) [12,13,14].

**Figure 2 pharmaceutics-14-00340-f002:**
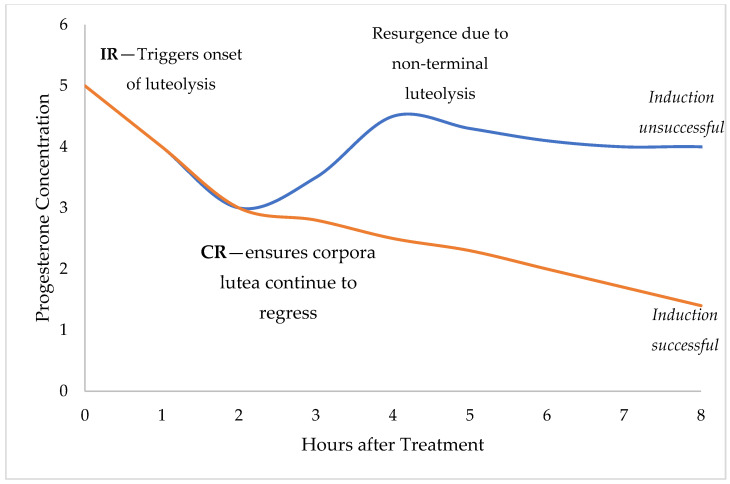
Graphical representation of how progesterone concentrations in sows are predicted to respond to the IR and CR formulations. An ideal RR NVDDS would trigger onset of luteolysis (orange and blue line). Administering the CR NVDDS should release cloprostenol gradually, triggering complete regression and result in a successful induction (orange line). Without the CR NVDDS, risk of resurgence due to non-terminal luteolysis could result in an unsuccessful induction (blue line).

**Figure 3 pharmaceutics-14-00340-f003:**
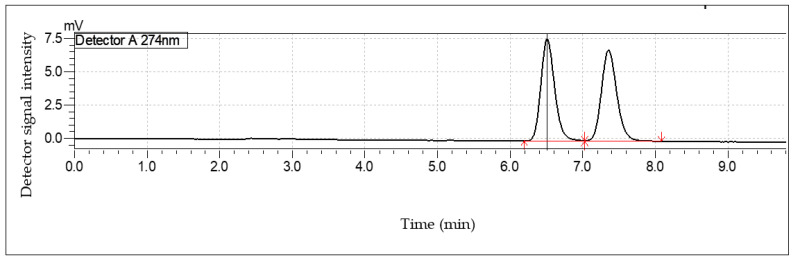
HPLC chromatogram of (±)-cloprostenol was separated into its two enantiomers using the method outlined by Kalikova et al. [22].

**Figure 4 pharmaceutics-14-00340-f004:**
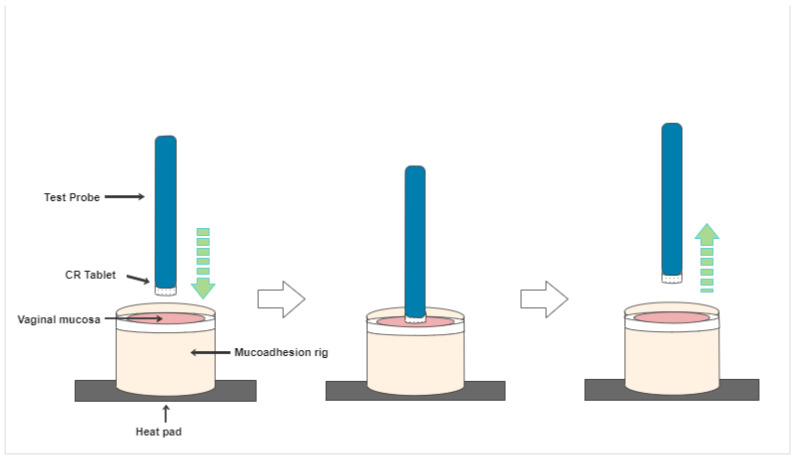
TA.XTplus texture analyser set up. The vaginal mucosa was cut to appropriate size to fit mucoadhesion rig support ring and the probe. Prior to testing, the vaginal mucosa was hydrated in PSVF for 15 min at 37 ± 1 °C.

**Figure 5 pharmaceutics-14-00340-f005:**
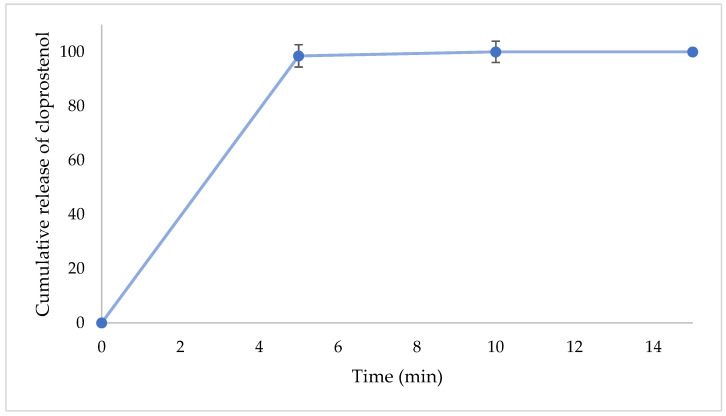
In vitro release profile of cloprostenol IR formulation. Each data point is the mean ± standard deviation of the triplicate.

**Figure 6 pharmaceutics-14-00340-f006:**
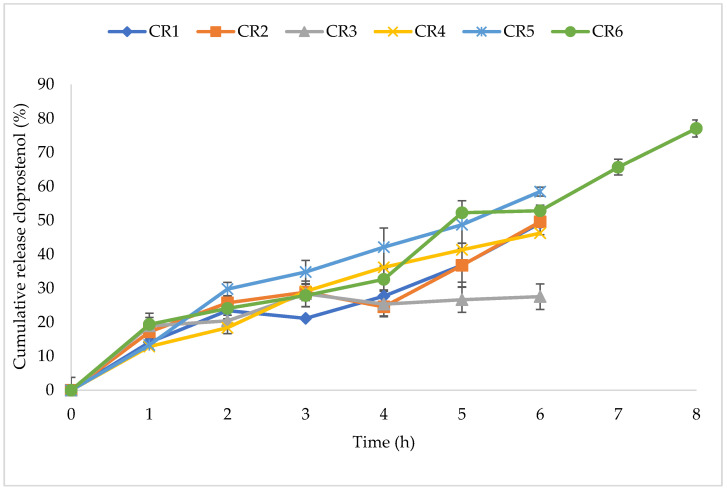
In vitro release profiles of cloprostenol CR NVDDS formulations. Each data point is the mean ± standard deviation of the triplicate. Formulation CR6 displayed favourable release and was assessed over an eight-hour time period.

**Figure 7 pharmaceutics-14-00340-f007:**
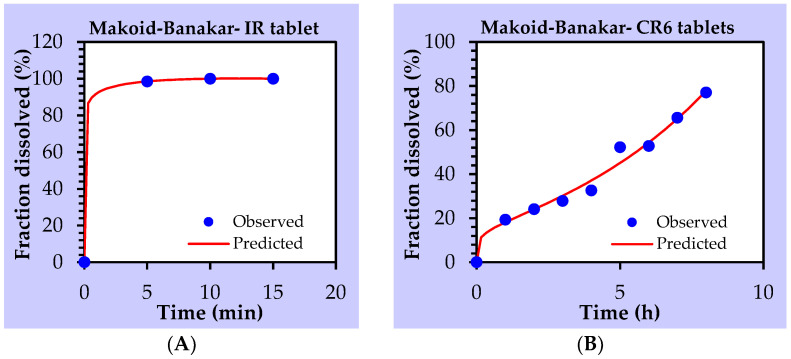
Release profiles of IR tablet (**A**) and CR6 tablet (**B**) fitted by Makoid–Banakar model.

**Figure 8 pharmaceutics-14-00340-f008:**
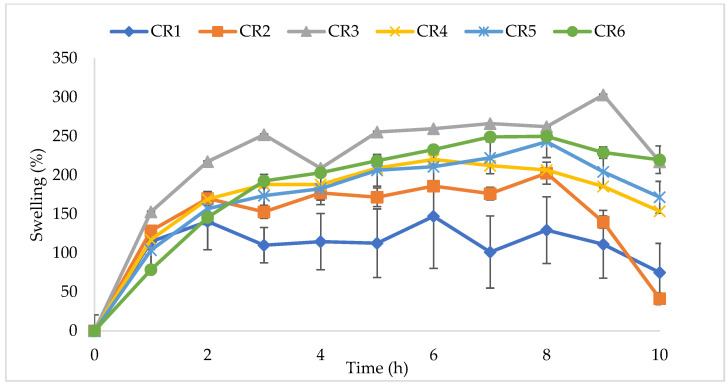
Swelling profiles for controlled release formulations CR1–CR6. Each data point is the mean ± standard deviation of the triplicate.

**Figure 9 pharmaceutics-14-00340-f009:**
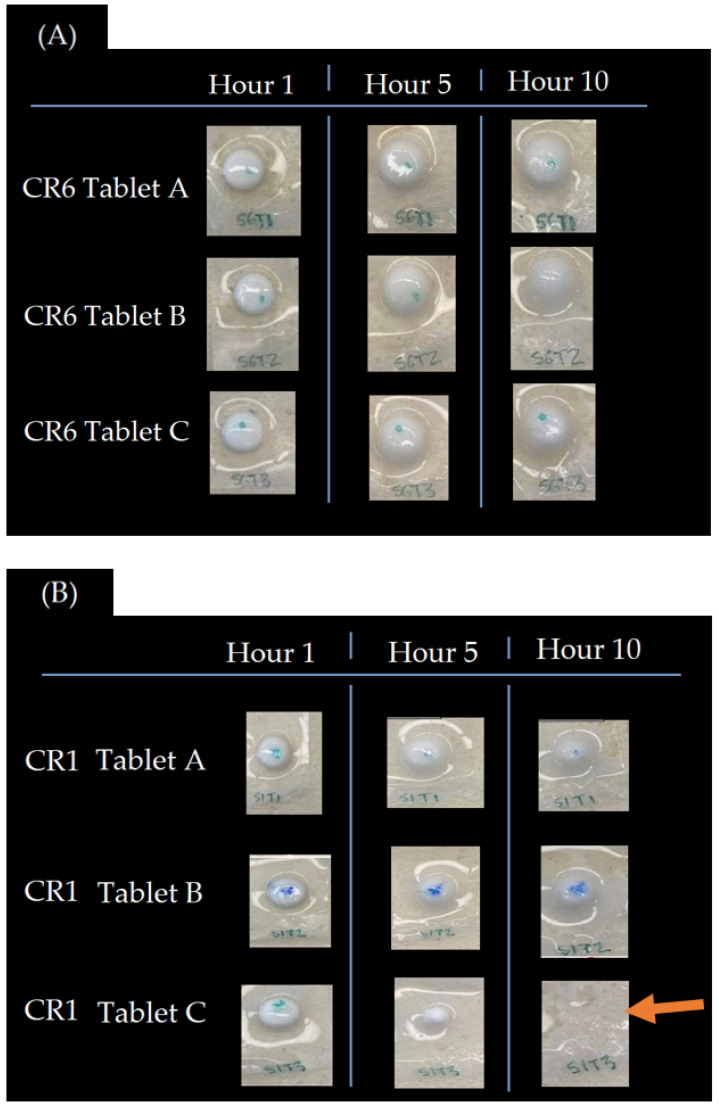
Formulation CR6 (**A**) and CR1 (**B**) after 1, 5 and 10 h of being submerged in PSVF. Uneven tablet erosion is evident between CR1 triplicates (**B**), as indicated by the arrow.

**Figure 10 pharmaceutics-14-00340-f010:**
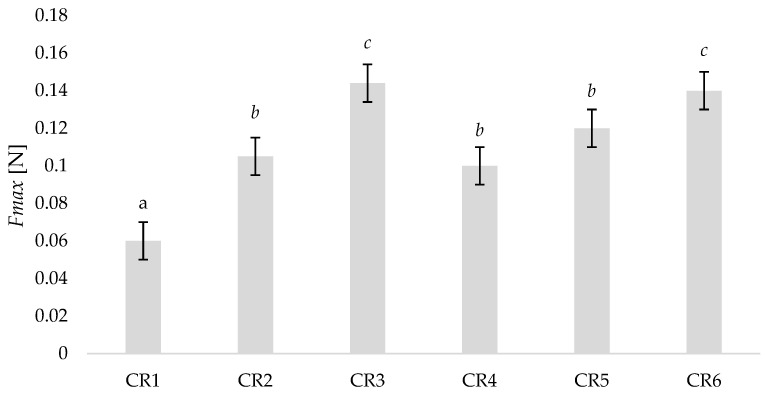
Mean detachment force Fmax (Newtons) for controlled release formulations CR1–CR6 as measured by a TA.XTplus texture analyser (*n* = 3). Formulations with significant differences in mean detachment force (at the 95% level of confidence; *p*-value < 0.05) are represented by different superscripts (*a*, *b* or *c*).

**Figure 11 pharmaceutics-14-00340-f011:**
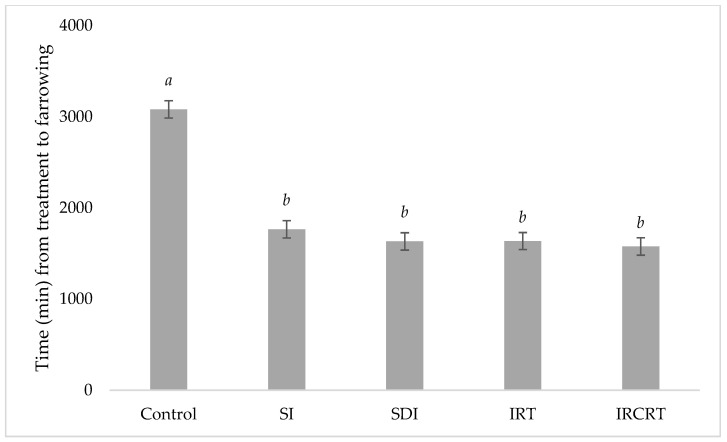
Time taken (minutes) for sows to farrow from time of treatment application to birth of the first piglet. The sows were assigned to five groups. Group 1 (SI) received a single vulval injection of cloprostenol at 0700 h (*n* = 32), group 2 (SDI) received the same dose split in two parts, at 0700 and 1300 h (*n* = 33). Group 3 (IRT) animals were administered an IR tablet at 0700 h (*n* = 32), while group 4 (IRCRT) received both IR and CR tablets at 0700 h (*n* = 33). Group 5 was untreated and served as a control (*n* = 32). Formulations with significant differences in mean detachment force (at the 95% level of confidence; *p*-value < 0.05) are represented by different superscripts (*a, b*).

**Table 1 pharmaceutics-14-00340-t001:** Composition of each tablet in controlled-release formulations (CR1–CR6). Hypermellose was obtained from the Dow Chemical Company as METHOCELL K100 Premium CR, K15M Premium CR, K4M Premium CR, and E50 Premium.

Formulation	Lactose (mg)	HPMC K100 (mg)	HPMC K15 (mg)	HPMC K4 (mg)	HPMC E50 (mg)	HPMC % (*w*/*w*)	Lactose % (*w*/*w*)
CR1	85	20	-	-	-	10	46
CR2	85	15	5	-	-	10	46
CR3	60	-	15	30	-	24	32
CR4	55	-	-	50		27	29
CR5	60	-	20	-	20	23	32
CR6	55	-	-	30	20	26	29

**Table 2 pharmaceutics-14-00340-t002:** Physical properties of tablets for the immediate release and controlled-release formulations (CR1–CR6) (*n* = 10) ± standard deviation.

Formulation	Thickness (cm)	Hardness (kgf)
IR	2.54 ± 0.02	4.39 ± 0.36
CR1	2.44 ± 0.03	12.96± 1.21
CR2	2.47 ± 0.02	13.25 ± 1.41
CR3	2.53 ± 0.03	13.96 ± 0.47
CR4	2.56 ± 0.27	13.67 ± 0.37
CR5	2.55 ± 0.031	13.24 ± 0.37
CR6	2.55 ± 0.022	13.74 ± 0.40

**Table 3 pharmaceutics-14-00340-t003:** Testing selected formulations for tablet uniformity (*n* = 10).

NVDDS Formulation	Desired Drug Content (µg)	Average Drug Content (µg)	Standard Deviation	Uniformity Value Score (L1 ≤ 15)
Immediate release (IR)	125	127.4	4.87	10.45
Controlled release Formulation 6 (CR6)	125	126.7	6.52	13.92

**Table 4 pharmaceutics-14-00340-t004:** Testing selected formulations for tablet friability (*n* = 35).

NVDDS Formulation	Original Weight (g)	Weight after 100 Rotations (g)	Percentage Lost (%) ≤ 1%
Immediate release (IR)	6.591	6.586	0.075
Controlled release formulation (CR6)	6.503	6.495	0.123

**Table 5 pharmaceutics-14-00340-t005:** Effects on sow performance of cloprostenol administered either as a single vulval injection at 0700 h, vulval injection at both 0700 and 1300 h, insertion of a rapid release (IR) NVDDS at 0700 and 1300 h, or both IR and controlled release (CR6) tablets at 0700 h or untreated controls. The statistical significance of mean values is presented as a *p*-value, where *p* ≤ 0.05 indicates a significant result at the 95% level of confidence.

Measurements on Sow Litter Performance	Control	SI	SDI	IRT	IRCRT	*p*-Value
Parity	2.63 ± 1.6	2.56 ± 1.6	2.25 ± 1.6	2.30 ± 1.6	2.03 ± 1.6	0.585
Total born	11.94 ± 0.5	12.47 ± 0.6	13.38 ± 0.7	12.48 ± 0.3	12.6 ± 0.2	0.486
Born alive	11.25 ± 0.4	11.66 ± 0.5	12.28 ± 0.6	11.73 ± 0.3	11.94 ± 0.4	0.730
Birthweight (kg)	1.38 ± 0.05	1.44 ± 0.04	1.28 ± 0.05	1.36 ± 0.05	1.30 ± 0.02	0.272
Preweaning survival (%)	90.9 ± 1.99	91.43 ± 1.58	91.61 ± 1.51	90.04 ± 2.67	89.89 ± 2.05	0.962

## Data Availability

The data presented in this study are available on request from the corresponding author.

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
