# Peer review of "Development of a Novel Vaginal Drug Delivery System to Control Time of Farrowing and Allow Supervision of Piglet Delivery"

_pharmaceutics, 2022, doi:10.3390/pharmaceutics14020340_

Round 1

Reviewer 1 Report

This manuscript describes IR and CR vaginal tablets to control time of farrowing in sows. The overall merit of the manuscript is above average. However, the manuscript contains numerous errors (see my attached pdf) that need to be corrected before I will recommend that it be accepted for publication. 

Author Response

Thankyou for your review, please see attached document with amendments 

Reviewer 2 Report

Lines14-16: Review sentence for clarity

Line 22: Change “In vivo” to “in vivo”

Line 24: Clarify the IR tablet was administered at two time points

Line 65 and 68: Should “RR” be “IR”?

Line 94: Check sentence for clarity

Line 119: Should this be Table 1?

Line 274: Capitalize “makoid”

Line 303: Define “a, b, c” in Figure 10

Line 328: Change “suppository” to tablet

Line 332: Define “ a and b” in Figure 11

Line 395: Often when administering a drug via the vaginal mucosa, systemic drug levels will be low. This can be considered a positive that systemic levels are low.

It would be worthwhile to include more information on the current treatments for inducing partition and the drawbacks to the current standard.

Were any vaginal irritation studies conducted with the tables? The rabbit model could be used to assess vaginal irritation and also per PK. It would be a way to demonstrate low systemic exposure in plasma, but measurable local tissue concentrations.

Author Response

Thankyou for your edits, please see attached document for amendments and responses

Round 2

Reviewer 1 Report

The authors did a nice job of revising the manuscript.

However, I have two comments on the latest draft:

Line 16: Why suppositories? You developed tablets.

Line 47: The text in Figure 1 is out of focus. It needs to be revised.

Author Response

Thankyou, line 16 and Figure 1 has been updated

Reviewer 2 Report

Thank you for addressing each of my comments. Everything was appropriately revised as needed. 

Author Response

Thankyou again, updated version has been submitted